# Selegiline Modulates Lipid Metabolism by Activating AMPK Pathways of Epididymal White Adipose Tissues in HFD-Fed Obese Mice

**DOI:** 10.3390/pharmaceutics15112539

**Published:** 2023-10-27

**Authors:** Hye-Young Joung, Jung-Mi Oh, Min-Suk Song, Young-Bae Kwon, Sungkun Chun

**Affiliations:** 1Department of Physiology, Jeonbuk National University Medical School, Jeonju 54907, Republic of Korea; hyjyj007@hanmail.net (H.-Y.J.); biojmi@jbnu.ac.kr (J.-M.O.); 2Research Institute for Endocrine Sciences, Jeonbuk National University Medical School, Jeonju 54907, Republic of Korea; 3Department of Microbiology, Chungbuk National University College of Medicine and Medical Research Institute, Cheongju 28644, Republic of Korea; songminsuk@chungbuk.ac.kr; 4Department of Pharmacology, Jeonbuk National University Medical School, Jeonju 54907, Republic of Korea; 1972y@jbnu.ac.kr

**Keywords:** selegiline, anti-obesity, adipogenesis, lipogenesis, lipolysis, eWAT browning

## Abstract

Obesity, as a major cause of many chronic diseases such as diabetes, cardiovascular disease, and cancer, is among the most serious health problems. Increased monoamine oxidase (MAO) activity has been observed in the adipose tissue of obese humans and animals. Although previous studies have already demonstrated the potential of MAO-B inhibitors as a treatment for this condition, the mechanism of their effect has been insufficiently elucidated. In this study, we investigated the anti-obesity effect of selegiline, a selective MAO-B inhibitor, using in vivo animal models. The effect was evaluated through an assessment of body energy homeostasis, glucose tolerance tests, and biochemical analysis. Pharmacological inhibition of MAO-B by selegiline was observed to reduce body weight and fat accumulation, and improved glucose metabolism without a corresponding change in food intake, in HFD-fed obese mice. We also observed that both the expression of adipogenenic markers, including C/EBPα and FABP4, and lipogenic markers such as pACC were significantly reduced in epididymal white adipose tissues (eWATs). Conversely, increased expression of lipolytic markers such as ATGL and pHSL and AMPK phosphorylation were noted. Treating obese mice with selegiline significantly increased expression levels of UCP1 and promoted eWAT browning, indicating increased energy expenditure. These results suggest that selegiline, by inhibiting MAO-B activity, is a potential anti-obesity treatment.

## 1. Introduction

Obesity is a major risk factor for metabolic diseases, including type 2 diabetes mellitus, inflammation, and cardiovascular disease [1,2]. Obesity occurs when energy intake is greater than energy expenditure. Increasing energy expenditure is therefore a potential therapeutic strategy for the treatment of obesity [1,3].

Adipose tissue is involved in the regulation of whole-body energy metabolism, and is responsible for breaking down lipid droplets into free fatty acids (substrates for energy metabolism) as well as lipid storage through adipogenesis. This imbalance in energy metabolism regulation leads to obesity, where excess energy is stored in fat cells, causing hyperplasia (increase in cell number) and hypertrophy (increase in cell size) [4,5]. For that reason, it is considered important to develop anti-obesity drugs to properly understand and control the regulatory mechanisms of adipogenesis and lipogenesis [6].

Thermogenesis generally occurs in brown adipose tissue (BAT), but in adipocytes thermogenesis is also possible in brown-like white adipocytes (also known as beige adipocytes). The process by which adipocytes acquire the characteristics of BATs is referred to as browning, and mouse studies show a correlation between this process and improvements to metabolic disease [7]. Achieving a positive energy balance by inhibiting excessive white adipose tissue (WAT) deposition, while stimulating WAT browning and activating BAT thermogenesis are considered as potential therapeutic targets for the treatment of obesity.

Monoamine oxidases (MAOs) are widely distributed enzymes that catalyze oxidative deamination of biogenic amines involved adrenalin, dopamine, and serotonin to produce hydrogen peroxide. Previous studies of MAO enzymes showed they are also expressed in the adipocytes of humans [8,9] and rodents [10,11]. These enzymes have been shown to be increased in the differentiated 3T3-L1 and white adipocytes of obese animal models [11,12]. MAO also accelerates adipogenesis by generating reactive oxygen species (ROS) [13]. Furthermore, the MAO inhibitors not only stimulate glucose uptake [14] but also inhibit lipolysis [15] in the adipocytes of rats.

Selegiline, a selective MAO type B inhibitor, effectively reduced obesity induced by a high-fat and high-sugar (HFS) diet in rats. It also played a role in reducing adipose tissue metabolism and inflammation. However, it did not affect body weight gain, impaired glucose homeostasis, or behavior [16]. Previous research suggests that the combined administration of MAO and semicarbazide-sensitive amine oxidases (SSAOs) can help alleviate fat accumulation and reduce body weight in obese Zucker rats [17,18]. Pargyline, a monoamine oxidase inhibitor administered at a dosage of 30 mg/kg, promoted lipolysis and increased the levels of free fatty acids in rats [19]. A recent study reported that selegiline demonstrated a protective effect against HFD-induced dyslipidemia and hepatic steatosis [20]. Despite the established metabolic benefits of selegiline in obese rodents, the specific mechanisms behind MAOB inhibition and lipid metabolism remain unclear. Therefore, this study aims to investigate the effects of selegiline on obese mice fed an HFD and elucidate any potential underlying mechanisms.

In our research, we have demonstrated that selegiline influences lipid metabolism and induces the browning of white adipocytes by activating AMP-activated protein kinase (AMPK). This identified molecular signaling pathway in eWAT may explain the anti-obesity effects of selegiline in vivo.

## 2. Materials and Methods

### 2.1. Animal Experiments

In vivo animal experiments were performed consistent with the guidelines established by the Jeonbuk National University Institutional Animal Care and Use Committee (IACUC) (Approval number: JBNU 2022-026). C57BL/6 mice (male, 6 weeks old) were purchased from Nara Biotech (Kyunggi-do, Korea) and housed with a 12 h light/dark cycle (light on, 08:00). The animals were allowed to adapt to laboratory conditions for a minimum of 7 days prior to the experiment. After becoming acclimated, mice were randomly assigned to either cohort 1 or cohort 2 (Figure 1). All mice were fed either a normal chow diet (NCD; 18% kcal fat, 3.1 kcal/g) or a high-fat diet (HFD; 60% kcal fat, 5.24 kcal/g; Research Diets, NJ, USA) and provided with ad libitum access to water.

Cohort 1: Mice were randomly divided into four groups: NCD+Veh group (*n* = 5): administration with vehicle to NCD fed mice; NCD+Selegiline group (*n* = 5): administration with selegiline (30 mg/kg) to NCD fed mice; HFD+Veh group (*n* = 5): administration with vehicle to HFD-fed mice; HFD+Selegiline group (*n* = 5): administration with selegiline (30 mg/kg) to HFD-fed mice. After 3 weeks of treatment, eWAT was collected and Hematoxylin and Eosin (H&E) staining was performed.

Cohort 2: A total of 20 mice were randomly divided into two groups: those fed an NCD (18% kcal fat, 3.1 kcal/g) and those fed an HFD (60% kcal fat, 5.24 kcal/g, Research Diets, NJ, USA). Food was provided for 9 weeks to establish obese mice models. The 20 mice were then weight matched and divided further into four groups: (1) NCD+Veh group (*n* = 5): administration with vehicle to NCD fed mice, (2) NCD+Selegiline group (*n* = 5): administration with selegiline (30 mg/kg) to NCD fed mice, (3) HFD+Veh group (*n* = 5): administration with vehicle to HFD-fed mice, (4) HFD+Selegiline group (*n* = 5): administration with selegiline (30 mg/kg) to HFD-fed mice. The selegiline was administered intraperitoneally once per day over 4 weeks. During the animal experiments, body weight and food intake were measured every day. At the end of the experiments, adipose tissue [eWAT, inguinal white adipose tissue (iWAT), BAT] was collected and subjected to H&E staining, immunohistochemistry (IHC) and Western blot analyses.

### 2.2. Glucose Tolerance Test (GTT)

The GTT was performed by modifying a previously reported method [21]. At the end of administration vehicle or selegiline, mice were fasted for 15 h (18:00 p.m.–9:00 a.m.) before GTT. The mice received an intraperitoneally injection of glucose (2 g/kg body weight, G8270, Sigma-Aldrich, St. Louis, MO, USA) and then, blood samples were collected from the tail at 0, 15, 30, 45, 60, 90, and 120 min to measure blood glucose concentration using a CareSensII Plus glucose meter (I-SENS Inc., Seoul, Korea). The area under the curve (AUC) was calculated using GraphPad Prism software, version 8.0.1 (GraphPad Software, San Diego, CA, USA).

### 2.3. Western Blotting Analysis

A Western blot was conducted as previously described [22]. Adipose tissues were homogenized in RIPA buffer with protease and a phosphatase inhibitor cocktail. Lysates were resolved on 8–10% sodium dodecyl sulfate-polyacrylamide gel and then transferred onto polyvinylidene fluoride membrane. The membranes were blocked with 5% skim milk in 0.1% Tween 20/Tris-buffered saline (TBST) and primary antibodies were incubated overnight at 4 °C using acetyl-CoA carboxylase (ACC: # (3662), phospho (p)ACC (#3661), AMPKα (#2532), p-AMPKα (#50081), CCAAT/enhancer-binding protein alpha (C/EBPα: #8178), fatty acid binding protein 4 (FABP4: #2120), peroxisome proliferator activated receptor gamma (PPARγ: #2435), Hormone-sensitive lipase (HSL: #4107), pHSL (#4137) (1:1000 dilution, Cell Signaling Technology, Beverly, MA, USA), Adipose triglyceride lipase (ATGL: sc-365278), PPARγ coactivator 1-alpha (PGC1α: sc-518025) (1:200 dilution, Santa Cruz Biotechnology Inc., Santa Cruz, CA, USA), PR/SET Domain 16 (PRDM16: Ab106410), Uncoupling Protein 1 (UCP1: ab155117) (1:1000 dilution, Abcam, Cambridge, UK), and Glyceralde-hyde-3-phosphate dehydrogenase (GAPDH: (AP0066; 1:5000, Bioworld Technology, Bloomington, MN, USA). After washing with TBST, membranes were incubated with anti-rabbit (#7074) or mouse (#7076) HRP-conjugated secondary antibody (1:5000, Cell Signaling Technology) for 1 h at room temperature. The expression levels of target proteins were quantified by image density scanning using ImageJ software Version 1.54d (NIH, Bethesda, MD, USA) and adjusted for GAPDH expression.

### 2.4. Hematoxylin and Eosin (H&E)

The paraffin-embedded epididymal adipose tissue sections (10 μm) were stained with Hematoxylin (S3309, Dako, Copenhagen, Denmark) and Eosin (HT110216, Sigma-Aldrich, St. Louis, MO, USA). The slides were mounted with mounting medium (ab104139; Abcam, Waltham, MA, USA). Images were captured using LAS software, version V4.9 (Leica, Hessen, Germany) under a light microscope (ICC50E, Leica) and adipocyte areas in the H&E stained cross-sections were analyzed using pixel-based AdipoArea software. The size of adipocytes was expressed as μm^2^.

### 2.5. Immunohistochemistry (IHC)

UCP1 immunofluorescence was performed using a method previously described in [23]. The tissue sections were placed for 10 min in a citrate buffer (pH of 6.0) at 60 °C for antigen retrieval and blocked with 5% normal goat serum/0.1 M PBS buffer. eWAT sections were incubated overnight in PBS at 4 °C with anti-UCP1 antibody (1:1000, Abcam) then further incubated in PBS for 2 h at room temperature with anti-rabbit IgG-Alexa 488 (ab150077, 1:1000, Abcam). After rinsing in PBS, the slides were mounted with Anti-Fade Fluorescence mounting medium (ab104135, Abcam). The images were acquired using a fluorescent microscope (CELENA S, Logos Biosystems, Anyang-si, Kyunggi-do, Korea).

### 2.6. Statistical Analysis

All data are presented as means ± SEM. Statistical analysis was performed using GraphPad Prism software version 8.0.1 (GraphPad Software, San Diego, CA, USA). The difference between groups was determined by one-way ANOVA followed by Dunnett’s multiple comparison test. Time course comparison between groups was analyzed using a two-way repeated measures (RM) ANOVA with Dunnett’s multiple comparisons. All *p* values < 0.05 were considered statistically significant.

## 3. Results

### 3.1. Selegiline Prevented Obesity in HFD-Fed Mice

We investigated if the processing of selegiline, a selective MAO-B inhibitor, is involved in body weight change in adipocytes from HFD-fed obese mice. Cohort 1, a group administered selegiline at the beginning of the trials, was established to elucidate the preventive effect of the MAO-B inhibitor on HFD-induced obesity. Selegiline (30 mg/kg, i.p.) was administered concurrently with feeding a normal chow or high-fat diet into mice (Figure 2A). We observed that the body weight and white fat pad weight (eWAT and iWAT) was significantly higher in the HFD+Veh group than in the NCD+Veh group (Figure 2B–D). Interestingly, simultaneous administration of selegiline over 3 weeks reduced body weight gain in HFD-fed mice. However, there was no change in body weight in the group fed regular food and the group administered selegiline in combination (Figure 2B,C). Additionally, simultaneous administration of HFD and selegiline decreased eWAT and iWAT weights compared to HFD mice, but there was no change in BAT (Figure 2D). We conducted H&E staining on eWAT and analyzed the impact of selegiline on adipocyte size change. Our experiment demonstrated that adipocytes in HFD mice were bigger than those in NCD mice. Co-administration of selegiline resulted in smaller adipocyte sizes in HFD mice compared to HFD alone (Figure 2E,F). Furthermore, based on a 3000 μm^2^ size distribution analysis of fat cells, the combined HFD and selegiline group had a greater increase in fat cell size below 3000 μm^2^ compared to just the HFD group, while the HFD group had a higher prevalence of fat cells larger than 3000 μm^2^ than the selegiline group (Figure 2G). In sum, these findings suggest that a combination treatment of selegiline during the early stages of HFD could effectively reduce the accumulation of WAT and body weight.

### 3.2. Selegiline Reduced Body Weight and Fat Accumulation in HFD-Fed Obese Mice

Selegiline was administrated to HFD-induced obese mice to confirm whether it was effective in treating the state of obesity. After being fed an NCD or HFD for 9 weeks, mice received intraperitoneal injections of selegiline (30 mg/kg) for 4 weeks (Figure 3A). Selegiline-treated mice exhibited significantly less body weight gain than mice in the HFD+Veh group (Figure 3B–D). We further examined whether selegiline could affect feeding behavior itself. As shown in Figure 3E, we observed that the daily food intake of mice fed an HFD was significantly higher than those fed an NCD. However, we observed no difference in daily food consumption amounts between selegiline-treated and untreated mice fed an HFD. This indicates that selegiline does not modulate food intake behavior. We next performed an intraperitoneal glucose tolerance test (GTT) to check for disturbances in glucose metabolism in all HFD mice. We found that selegiline treatment significantly improved the glucose tolerance of obese mice fed an HFD (Figure 3F). In sum, these results showed that the inhibition of MAO-B by the systemic administration of selegiline, while not changing the amount of food consumed by obese mice fed an HFD, nevertheless improved their glucose metabolism, thereby exerting a positive effect on the regulation of body weight and fat accumulation.

### 3.3. Selegiline Administration Reduced Epididymal Adipocyte Size

Comparing the size of adipocytes and the weight of adipose tissue is an important means of assessing the effectiveness of anti-obese agents. We examined whether selegiline treatment had any regulatory effect on the size of adipose cells or tissues by weighing eWATs and iWATs as well as BATs. As shown Figure 4A,B, selegiline treatment reduced fat mass in the eWATs and iWATs compared with the vehicle-treated HFD group, but the fat mass of the BAT did not change in either group. To investigate whether selegiline administration reduces adipocyte hypertrophy in HFD-induced obese mice, the size of eWATs, the primary component of WAT, was analyzed and compared between selegiline-treated and untreated groups. Tissue imaging analysis revealed that the eWATs were significantly smaller in the selegiline-treated groups than in the untreated groups (Figure 4C–F).

A more fine-grained analysis was performed to determine the mechanism by which the size of the adipocytes, such as eWAT, were reduced. Our experiments confirmed a change in the size distribution of adipocytes after treatment with selegiline; the number of small adipocytes (below 3000 μm^2^) was higher than that of large adipocytes (above 3000 μm^2^) post-treatment. This may be the cause of the overall decrease in body weight gain and fat mass that was observed to have followed selegiline treatment. Our results suggest that treatment of an obesity-induced mouse model with a MAO-B inhibitor can control the accumulation of adipocytes and exert an anti-obesity effect by regulating their distribution.

### 3.4. Selegiline Regulates Lipid Metabolism through an Activated AMPK Signaling Pathway in eWAT

To better understand the inhibitory effects of selegiline on fat accumulation in obese mice, we investigated the related molecular mechanisms for lipid metabolism. Protein levels of pAMPK, AMPK, PPARγ, C/EBPα, pACC, ACC, FABP4, ATGL, pHSL, and HSL were determined by Western blotting. Selegiline administration increased the ratio of p-AMPKα to AMPKα, indicating activation of the AMPKα signaling pathway (Figure 5A). In addition, selegiline significantly down-regulated the protein levels of adipogenic markers such as FABP4 and C/EBPα in the eWAT in obese mice, but slightly decreased PPARγ expression (though only to a degree that does not suggest statistical significance) (Figure 5A). Selegiline significantly increased phosphorylation of acetyl CoA carboxylase (pACC) by AMPK, indicating that the treatment inhibited the activity of ACC, a lipogenesis marker (Figure 5B). Phosphorylation of ATGL and hormone-sensitive lipase (HSL) was significantly higher in the selegiline-treated group than in the HFD-induced obese group, indicating that activated lipolysis occurred in the former. These results suggest that selegiline ameliorates HFD-induced obesity by inhibiting adipogenesis or lipogenesis and reducing fat pad storage through the activation of lipolysis.

Phosphorylation of ATGL and hormone-sensitive lipase (HSL) was significantly higher in the selegiline-treated group than in the HFD-induced obese group, indicating that activated lipolysis occurred in the former. These results suggest that selegiline ameliorates HFD-induced obesity by inhibiting adipogenesis or lipogenesis and reducing fat pad storage through the activation of lipolysis.

### 3.5. Selegiline Induced eWAT Browning of HFD-Fed Obese Mice

We further investigated whether eWAT browning was enhanced by selegiline treatment. We assessed this by quantifying PRDM16 and PGC1α protein levels, which are markers for brown-like adipocytes. As shown in Figure 6, PRDM16 and PGC1α protein expression was strongly induced by selegiline in eWATs from HFD mice, suggesting that the transition from white to brown-like adipocytes occurred. Treatment with selegiline was observed to have induced higher levels of UCP1 expression levels in the eWATs than in the untreated group (Figure 6A,B). This result was confirmed through fluorescence staining UCP1, which revealed more UCP1 positive staining in the HFD+Selegiline group than in the HFD-Veh group (Figure 6C). We did not have any significant difference in UCP1 expression levels in the BATs of the HFD-Veh and HFD+Selegiline groups (Figure 6D,E). Our results ultimately strongly suggest that the MAO-B inhibitor selegiline promotes the formation of brown or beige-like eWATs while simultaneously increasing thermogenesis in HFD-induced obese mice.

## 4. Discussion

In this study, we investigated the anti-obesity impact of selegiline on eWAT lipid metabolism and the underlying molecular mechanism in an obese mouse model given HFD. Our findings indicate that co-administration of selegiline and HFD reduces body weight and fat accumulation in WAT and also decreases the size of eWAT. Selegiline treatment in an animal model, where obesity was induced by HFD, reduced the weight of obese mice, white fat accumulation, and eWAT size. Furthermore, our study showed that selegiline improved glucose homeostasis, lipid metabolism, and promoted browning of WAT.

MAO activity is found not only in the brain but also in peripheral organs such as adipocytes [24]. In adipose tissue, MAO is highly expressed in mature adipocytes and appears during adipocyte differentiation [9]. Previous studies have identified selegiline as responsible for improved energy metabolism and suppressed inflammation in the adipose tissues of high-fat and high-sugar diet obese rats, but observed no change to body weight [16]. Other studies concerning the effect of a selective/non-selective MAO-inhibitor have shown that MAO inhibitors repress adipogenesis in hBM-MSCs [18,25]. Also, selegiline improved lipid metabolism in the liver of HFD-fed mice due to regulating fatty acid oxidation [20]. Another MAO inhibitor, phenelzine, has been reported to improve obesity-related complications [26]. These research results showed the potential for anti-obesity effects by targeting adipocyte MAOs. Similar to previous findings, this study demonstrates that selegiline (30 mg/kg) has potential for anti-obesity prevention and treatment in a mouse model with obesity induced by HFD.

The HFD mouse model induces obesity and is commonly used to screen for anti-obesity compounds. The phenotype of mice changes due to the obesity induced by the HFD, leading to weight gain and altered expression of genes related to lipids. C/EBPα, PPARγ, and FABP4 are key regulators of adipocyte differentiation and have been reported to be highly expressed in HFD mice [27].

Activation of AMPK is essential in combatting obesity as it hinders lipid assimilation and suppresses adipocyte differentiation. AMPK phosphorylation inhibited adipogenesis and lipogenesis, and increased lipolysis and thermogenesis. AMPK is a cellular energy sensor and regulator of metabolic homeostasis, which controls lipid and glucose metabolisms [28,29,30,31]. The activation level of AMPK in HFD-induced obese mice is lower than that in normal mice. According to previous studies, AMPK activation in peripheral tissues (such as liver and adipose tissue) suppresses obesity, type 2 diabetes, and related metabolic disorders [32,33,34,35]. Thus, although limited to peripheral tissues, AMPK activators (such as metformin, which is provided as a treatment for diabetes), which directly regulate lipid metabolism and indirectly activate thermogenesis, are considered as potential therapeutic agents for obesity and metabolic diseases [28,34,36,37,38].

In mammals, ACC acts as a downstream signal of AMPK, which facilitates the conversion of acetyl-CoA to malonyl-CoA. AMPK inhibited ACC1 and ACC2 activity by phosphorylating ACC1 Ser79 and ACC2 Ser212 in mice [39]. In general, many anti-obesity compounds hinder lipid synthesis by phosphorylating AMPK or ACC, resulting in their effectiveness [40,41]. Changes in adipose tissue lipid metabolism significantly impact whole-body energy homeostasis [42,43].

We then tested the effect of selegiline on lipid metabolism in epididymal adipose tissue. Our data showed a significant reduction in the expression of adipogenic genes (C/EBPα, FABP4) and down-regulated lipogenesis by ACC to pACC in HFD-fed obese mice administered selegiline. Selegiline significantly increased ATGL expression and HSL phosphorylation in the epididymal adipose tissues of HFD obese mice, which in turn promoted lipolysis. WAT is known to contain HSL, and lipolysis is activated through the phosphorylation of this enzyme.

Lipolysis is the process of breaking down triglycerides (TGs) stored in adipose tissue into free fatty acids (FFAs) and glycerol. Lipolysis in the WAT of humans and rodents is regulated in a step-wise fashion by adipose triglyceride lipase (ATGL), hormone-sensitive lipase (HSL), and monoacylglycerol lipase (MAGL) [44]. During the development of obesity, adipose tissue expands tremendously and adipocyte size increases to neutralize and store nutritional overload. However, when the adipocytes are eventually unable to store excess lipids, two important pathological processes in adipose tissue are important for the development of metabolic diseases: adipose tissue inflammation and hypertrophy. Therefore, complete lipolysis in adipocytes seems to necessitate the coordinated activation of ATGL, HSL, and MAGL. The findings of this investigation indicate that selegiline inhibits the increased expression of lipid-related genes caused by a HFD and triggers the phosphorylation of AMPK, suggesting a possible link between AMPK phosphorylation and the anti-obesity effect of selegiline treatment.

UCP1 inhibits ATP production, and increased UCP1 in adipose tissue stimulates the breakdown of triglycerides stored as fat [45]. WAT plays an important role in maintaining systemic homeostasis [46]. Elevated UCP1 levels in WAT may be due to its conversion to BAT, which has thermogenic properties [45]. Several studies have shown that PRDM16 primarily regulates transcription factors such as PGC1α [47,48]. There is evidence that reducing the expression of PRDM16 reduces thermogenic properties in brown adipocytes [49,50]. Together, these proteins play an important role in regulating energy metabolism in adipocytes.

Our results revealed that selegiline stimulated the protein expression of UCP1 in the WAT of HFD-fed obese mice. Consistent with this upregulated expression, the expression of PGC1α and PRDM16 of the brown adipocyte marker was also upregulated by selegiline. In short, treatment with selegiline induced lipolysis and indirectly affected browning of white adipocyte and thermogenesis in the HFD-fed mice. Although we observed an increase in UCP1 expression in the eWATs, there was no change in UCP1 expression in the BATs, a major organ of thermogenesis. These outcomes implied that the activation of AMPKα, and the subsequent increased expression of brown adipocyte marker genes, might play a role in the selegiline-stimulated browning of eWAT in HFD obese mice.

Despite showing the anti-obesity effect of selegiline, our study has limitations. Selegiline is clinically administrated orally to depression patients (5~10 mg/day) [51]. In previous studies, obese animals were subcutaneous administered dose of 0.25~0.6 mg/kg and 10 mg/kg daily dissolving the compound in drinking water [52], it is not known whether this dose is optimal for treatment of obese. Furthermore, our study is limited by using a single dose of 30 mg/kg (roughly 1 mg/day). And, using only an obese animal model, we were unable to conduct experiments on the anti-obesity effect of MAO-B inhibition by directly controlling the brain–adipose tissue axis, and by observing changes in dietary amount, there was no change in eating behavior.

Our study demonstrated the mechanism of action of selegiline, an MAO-B inhibitor, in HFD-induced obese mice. Selegiline inhibits adipogenesis through AMPK activation and exhibits preventive and therapeutic effects by inducing lipolysis in adipose tissue, reducing adipogenesis markers (C/EBPα and FABP4), an adipogenesis marker (pACC), and lipolysis markers (ATGL and pHSL) were increased resulting in a reduction in fat accumulation in adipose tissue. In addition, selegiline-activated AMPK induced WAT browning by increasing UCP1 levels, which triggers energy expenditure using fatty acids as substrates. Throughout this study, we have provided evidence to support the use of selegiline and its mechanisms as an anti-obesity agent.

Collectively, the present study demonstrated that selegiline prevents HFD-induced obesity in vivo, as well as inhibits the expression of adipogenesis- and lipogenesis-related proteins and stimulates the expression of lipolysis-related proteins in the eWAT of HFD-fed mice. Our results further show that selegiline induces the expression of thermogenesis-related proteins and promotes browning in the eWAT of obese mice. In conclusion, epididymal adipocyte browning was induced via phosphorylated AMPK, meaning that it may contribute to the anti-obesity effects of selegiline in HFD-induced obese mice.

## Figures and Tables

**Figure 1 pharmaceutics-15-02539-f001:**
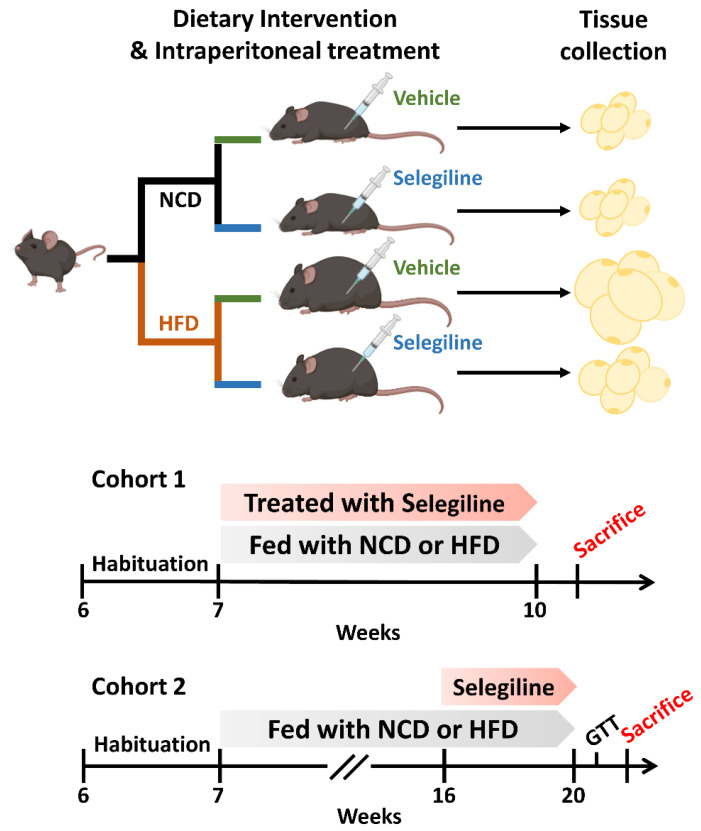
Schematic illustration of the experimental configuration.

**Figure 2 pharmaceutics-15-02539-f002:**
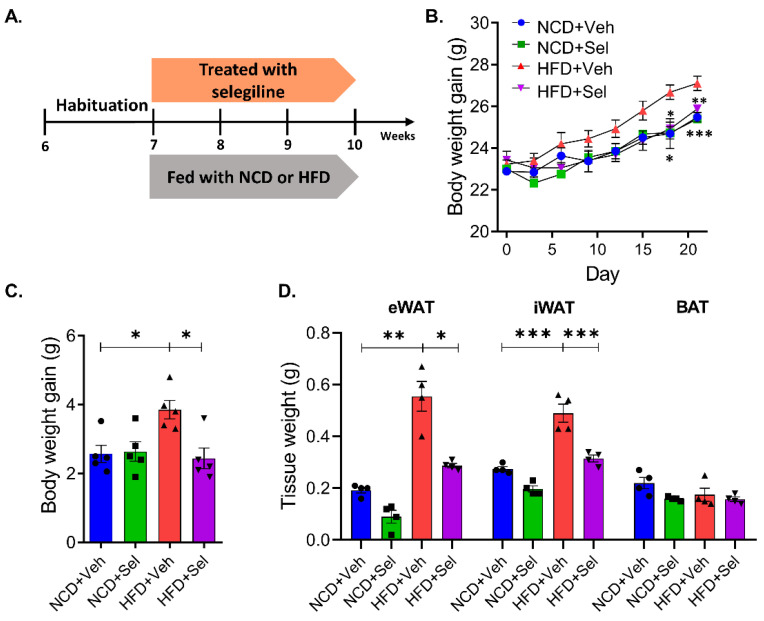
Selegiline prevented weight gain in HFD-fed mice. (**A**) Schematic diagram for this experiment. C57BL/6 mice were fed NCD or HFD and intraperitoneally treated with vehicle or selegiline (30 mg/kg) once a day for 3 weeks. (**B**) Body weight curves of vehicle- and selegiline-treated mice fed NCD or HFD (*n* = 5 per group). (**C**) Graph of body weight gain each groups. (**D**) Bar graph of weight of eWATs, iWATs, and BATs. (**E**) Representative of H&E staining of eWATs. Scale bar = 50 μm. (**F**) Summary plot of average of adipocyte areas. (**G**) Summary plot of frequency of large adipocytes, defined as having an adipocyte area greater than 3000 μm^2^. Data are presented as the mean ± SEM and one-way ANOVA followed by Dunnett’s multiple comparison test was employed for data analysis. Significance denoted by *: *p* < 0.05, **: *p* < 0.01, ***: *p* < 0.001 compared to HFD+Veh. NCD, normal chow diet; HFD, high-fat diet; Veh, vehicle; Sel, selegiline; eWAT, epididymal white adipose tissues; iWAT, inguinal white adipose tissue; BAT, brown adipose tissue.

**Figure 3 pharmaceutics-15-02539-f003:**
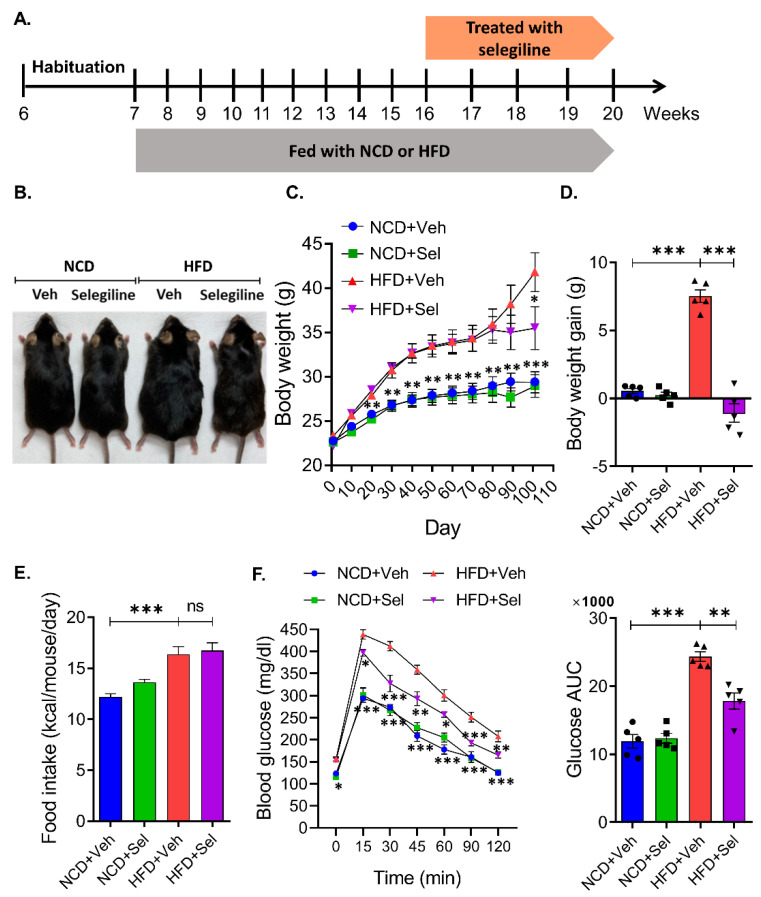
Selegiline markedly reduced body weight and fat accumulation in HFD-fed obese mice. (**A**) Schematic diagram for this experiment. C57BL/6 mice were fed an NCD or HFD for 9 weeks, and then intraperitoneal administrated 30 mg/kg of selegiline or saline once a day for 4 weeks. (**B**) Representative figure of body weight change in each group. (**C**) Polled data of body weight of each group. One-way ANOVA followed by Dunnett’s multiple comparison test. (**D**) Graph of body weight gain by each group. One-way ANOVA followed by Dunnett’s multiple comparison test. (**E**) Food intake of each group. (**F**) Left: Polled data of GTT of each group. Two-way RM ANOVA followed by Dunnett’s multiple comparison test, interaction F(6,108) = 5.398, *p* < 0.0001, time F(6,108) = 351.8, *p* < 0.0001, between group F(3,18) = 71.06, *p* < 0.0001. Right: Bar graphs showing area under the curve (AUC) values obtained from GTT experiments. Data are presented as the mean ± SEM. Significance was denoted by *: *p* < 0.05, **: *p* < 0.01, ***: *p* < 0.001 compared to HFD+Veh. NCD, normal chow diet; HFD, high-fat diet; Veh, vehicle; Sel, selegiline.

**Figure 4 pharmaceutics-15-02539-f004:**
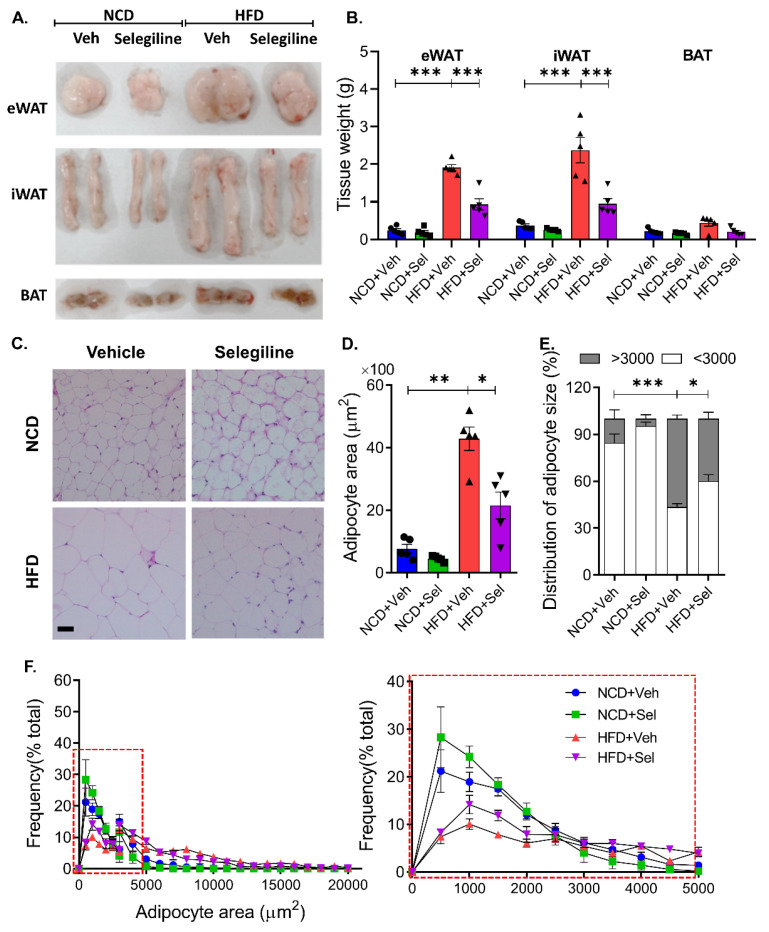
Selegiline administration reduced epididymal adipocyte size in HFD-fed obese mice. (**A**) Representative figure of eWATs, iWATs, and BATs. (**B**) Polled data of weight of eWATs, iWATs, and BATs. (**C**) Representative images of H&E staining of eWATs from each group. Scale bar = 50 μm. (**D**) Average of adipocyte size of WAT were measured from H&E images using AdipoArea software. *n* = 5 per group. (**E**) Frequency of large adipocytes, defined as having an adipocyte area greater than 3000 μm^2^. (**F**) Summary plot of quantitative analysis of adipocyte area. Data are presented as the mean ± SEM and one-way ANOVA followed by Dunnett’s multiple comparison test was employed for data analysis. Significance denoted by *: *p* < 0.05, **: *p* < 0.01, ***: *p* < 0.001 compared to HFD+Veh. NCD, normal chow diet; HFD, high fat diet; Veh (V), vehicle; Sel, selegiline; eWAT, epididymal white adipose tissues; iWAT, inguinal white adipose tissue; BAT, brown adipose tissue.

**Figure 5 pharmaceutics-15-02539-f005:**
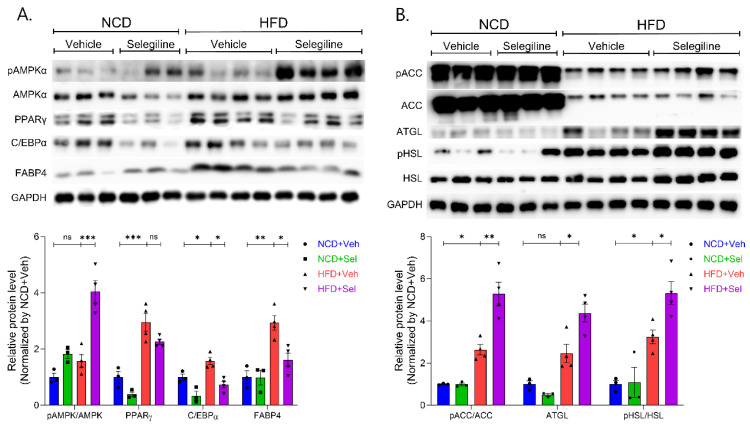
Selegiline-regulated lipid metabolism in epididymal WATs. (**A**) Upper panel: Images of Western blot of pAMPK, AMPK, PPARγ, C/EBPα, FABP4, and GAPDH in WATs from mice treated with selegiline or vehicle fed an NCD or HFD. Lower panel: Summary plot of expression of ratio of pAMPK/AMPK, PPARγ, C/EBPα, and FABP4. (**B**) Upper panel: Images of Western blot of pACC, ACC, ATGL, pHSL, HSL, and GAPDH in eWATs from mice treated with selegiline or vehicle fed an NCD or HFD. Lower panel: Summary plot of expression of ration of pACC/ACC, ATGL, ratio of pHSL/HSL. Data are presented as the mean ± SEM and one-way ANOVA followed by by Dunnett’s multiple comparison test was employed for data analysis. Significance denoted by *: *p* < 0.05, **: *p* < 0.01, ***: *p* < 0.001 compared to HFD+Veh. p(Phospho)AMPKα, AMP-activated protein kinase α; PPARγ, Peroxisome proliferator activated receptor gamma; C/EBPα, CCAAT/enhancer-binding protein alpha; FABP4, fatty acid binding protein 4; GAPDH, Glyceraldehyde-3-phosphate dehydrogenase; pACC, Phospho-Acetyl-CoA Carboxylase; ATGL, Adipose triglyceride lipase; HSL, Hormone-sensitive lipase.

**Figure 6 pharmaceutics-15-02539-f006:**
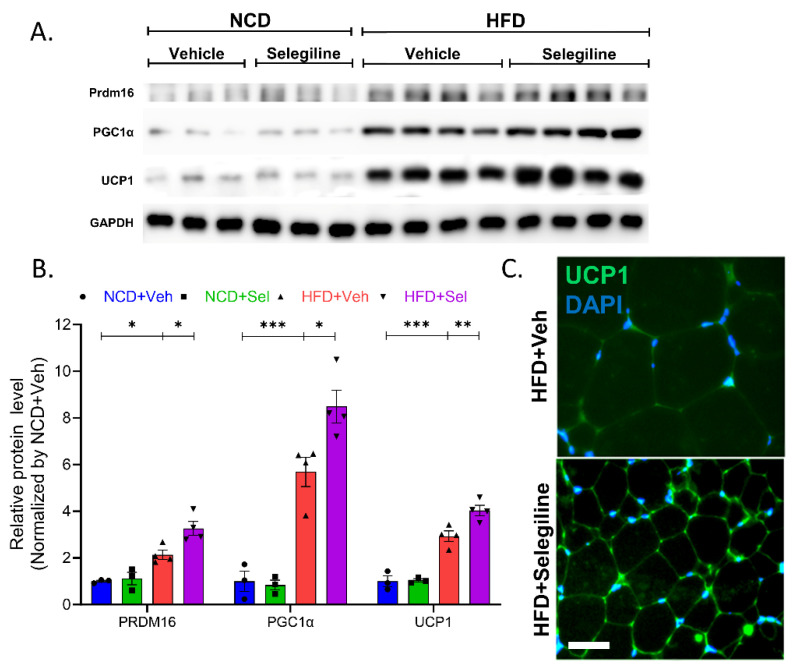
Selegiline promoted thermogenesis and epididymal WAT browning in HFD-fed obese mice. (**A**) Images of Western blot of PRDM16, PGC1α, UCP1, and GAPDH in eWAT from mice treated with selegiline or vehicle fed an NCD or HFD. (**B**) Summary plot of expression of PRDM16, PGC1α, and UCP1 in eWATs from mice treated with selegiline or vehicle fed an NCD or HFD. (**C**) Representative of UCP1 immunostaining of eWAT. Scale bar = 50 μm. (**D**) Images of Western blot UCP1 and GAPDH in BATs from mice treated with selegiline or vehicle fed an NCD or HFD. (**E**) Summary plot of expression UCP1 in BATs from mice treated with selegiline or vehicle fed an NCD or HFD. Data are presented as the mean ± SEM and one-way ANOVA followed by Dunnett’s multiple comparison test was employed for data analysis. Significance denoted by *: *p* < 0.05, **: *p* < 0.01, ***: *p* < 0.001 compared to HFD+Veh. Prdm16, PR/SET Domain 16; PGC1α, Peroxisome proliferator-activated receptor gamma coactivator 1-alpha; UCP1, Uncoupling Protein 1; GAPDH, Glyceraldehyde-3-phosphate dehydrogenase; NCD, normal chow diet; HFD, high-fat diet; Veh, vehicle; Sel, selegiline; BAT, brown adipose tissue.

## Data Availability

All data is contained within this article. Further inquiries can be directed to the corresponding authors.

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
