# Peer review of "Selegiline Modulates Lipid Metabolism by Activating AMPK Pathways of Epididymal White Adipose Tissues in HFD-Fed Obese Mice"

_pharmaceutics, 2023, doi:10.3390/pharmaceutics15112539_

Round 1
Reviewer 1 Report
Comments and Suggestions for Authors
The current study examined the anti-obesity effect of Selegiline in vivo using HFD-fed C57BL/6 mice, which has been reported before. The overall experimental design of this study is reasonable, the results are convincing and well-presented. The methods and results are overall well written. However, there are several critics that warrant further consideration of the acceptance of this manuscript as below:
1. The major concern of this study is the lack of novelty. A recent study (PMID: 36822098) by Tian et al. reported that “Selegiline ameliorated dyslipidemia and hepatic steatosis in high-fat diet mice”, which has similar design, results, and more mechanism study. However, the authors failed to even mention and cite this study in their manuscript. Another study (PMID: 29971762) also reported similar results using rat as model, which was cited by the authors. In the current study, although there are differences as compared to the above studies. There is not much further contribution to the mechanism of the anti-obesity effect of Selegiline. Using only in vivo mouse model is difficult to elucidate molecular mechanisms of function. Many parameters are simply endpoints resulting from or caused simply by reduced body weight.
2. Another critical question is the authors used “inhibition of MAO-B by selegiline” or “by inhibiting MAO-B” throughout the manuscript. It is not clear how a MAO-B inhibitor plays a role in obesity. The authors failed to discuss the association between a MAO-B inhibitor and obesity in the manuscript, including current research opinion and proposed mechanisms. Since dopamine level was not measured in the manuscript, the way the authors wrote suggests that the anti-obesity effect of Selegiline is based on its anti-MAO effect, which is inappropriate and misleading. Please refer to lines 22; 29; 153; 200; 232.
3. The discussion is fairly weak, which was a repeat of the results, lacking citations and explanations to interpret data.
4. Minor issues are:
(1) In 2.1 Animal experiments: number of mice were missing for several groups, line 83 and 84. The dose of Selegiline is missing in this section.
(2) How the dose of Selegiline was chosen, which was mentioned in the results section on line 159. Why did you use 30 mg/kg by IP, but in Tian’s paper, 0.6 mg/kg body weight was administered to mice by IP?
(3) Figure 1 cohort2 “Treated with selegiline” was placed in wrong place.
(4) On line 117, a citation is missing.
(5) In Figures 2 and 3, 2A and 3A are unnecessary and should be deleted.
(6) On line 175, what does 3000 um2 mean?
(7) In Figure 5, how do you explain why the expressions of pACC and ACC were much less upon HFD, but were highly expressed at normal diet?
(8) Similarly, in Figure 6, why HFD increased the expression of UCP-1 both in WAT and BAT?
Comments on the Quality of English LanguageOverall, well written, no much concern.
Author Response
Thanks for the reviewer's constructive comments. We attached our responses with a PDF file.

Reviewer 2 Report
Comments and Suggestions for Authors
The manuscript entitled “Selegiline modulates lipid metabolism by activating AMPK pathways of epididymal white adipose tissues in HFD-fed obese mouse” has elucidated the role of selegiline, MAO-B inhibitor, in regulation of eWAT hypertrophy and browning during the development of obese. The result was well written and shown a promising function of selegiline in lipid metabolism in eWAT. However, more experiments are needed to reach the conclusions “selegiline, by inhibiting MAO-B activity, is a potential anti-obesity treatment”, especially on the “link” from MAO-B to AMPK signaling pathway.
Major Concerns:
1, Logically, lacking of data connecting MAO-B to AMPK signaling and browning. Besides, there was missing of clear descriptions and discussions about the connection.
2, Considering MAO-B also expresses in hypothalamus, and it is known that the AMPK of hypothalamus is involved in eWAT browning, it is pity that the work didn’t provide data to show whether the underlying mechanism of eWAT browning was its targeting to hypothalamus or/and eWAT directly. Therefore, pharmacologically blocking the transmission from hypothalamus to eWAT or directly study the effect of selegiline on eWAT in vitro might be necessary.
3, Figure 4, iWAT was also decreased by selegiline, what’s the possible explanation?
4, Figure 5B, for ACC and pACC, please upload a weaker exposed image. N=3-4 was not enough for animal studies, please increase animal number.
5, ACC was strikingly downregulated by HFD in figure 5B, was that normal? what the possible reason?
6, for the previous study, the food intake was decreased in HFD mice (Selegiline reduces adiposity induced by high-fat, high-sucrose diet in male rats. British Journal of Pharmacology. https://doi.org/10.1111/bph.14437). Here, the food intake was increased, what’s the possible explanation?
Minor concerns:
1, please revise “NCD” by “Chow” in figure 1 and replace “IPGTT” by “IGTT” or “GTT” in the manuscript.
2, change the colors of these 4 groups to show the groups clearer.
3, figure 6B, missing the Y-axial labeling.
Comments on the Quality of English LanguageWell written.
Author Response

(The authors gave the same response as above.)

Reviewer 3 Report
Comments and Suggestions for Authors
This is an important original contribution on the potential therapeutic benefits of “Selegiline” against obesity related complications. The manuscript is of interest, however requires major revisions, to enhance quality and value to contributing to the scientific knowledge.
Specific comments
The use of a single dose of selegiline within this study is a major drawback. Why not also have a comparative dose? In fact, why was there no reference/comparative drug to compare the results?
The author should revise the abstract. The seen activity in terms of activity of the pharmacological compound could not be directly attributed to AMPK activation, unless they perform more (inhibition) studies. Thus, they should revise the statement, stating “…through the activation of AMPK pathways in epididymal white adipose tissues (eWATs)”
There are sections without proper citations within the document, or author forget to add citations. The document should be revised and precise citation should be added, including removal of words such as “(ref)” within texts.
Add citations, for the sentence, “Building on this prior work,…”
The methodology describing therapeutic intervention should clearly described. Do not add unnecessary citations, or rather define all abbreviations at first mention.
Provide motivation for selection of diets, experimental models, and doses used for the drug of interest. All this information should relevantly be clear to the reader.
Why use 4 animals per group? Did this not affect the statistical analysis?
Provide all resources for equipment’s and reagents used, which is essential for reproducibility of results.
Dilutions, and sources, including catalogue numbers should be given for antibodies used
Make figure captions more informative, to clearly indicate what is happening or being discussed within each figure. Also add relevant abbreviations within the caption, if used.
The use of GAPDH alone, as a reference gene (especially for PCR) is questionable. Also surrender the probes/primers used.
The authors cannot say “Selegiline prevented obesity in HFD-fed mouse” You have to refine this and be specific on what is being reported, whether its reducing body weights, adipocyte size or metabolic parameters in mice.
The discussion should also guide translation of findings, especially relevance of bioavailability or pharmacokinetics of the compound being tested, how does it affect clinical setting eventually?
The discussion should be enriched to discuss other relevant studies, comparing bioactive compounds/drugs, especially how do these compare to the effect seen with Selegiline
The study requires addition of limitations, or strengths, guiding future directions, drawbacks should also be included as part of the abstract
Comments on the Quality of English Language
The English Language is fine, but proof-reading is required to improve the flow and eliminate obvious mistakes.
Author Response

(The authors gave the same response as above.)

Reviewer 4 Report
Comments and Suggestions for Authors
Selegiline modulates lipid metabolism by activating AMPK pathways of epididymal white adipose tissues in HFD-fed obese mouse
This study showed the mechanism of selegiline, MAO-B inhibitor, against HFD-induced obese mice. Selegiline exerted preventive and curative effects that inhibited adipogenesis and induced lipolysis in adipose tissue through AMPK activation resulting in decreasing adipogenenic markers (C/EBP-ɑ and FABP4) and lipogenic marker (pACC), as well as increasing lipolytic markers (ATGL and pHSL), consequent to reducing fat accumulation in adipose tissue. Moreover, Selegiline-activated AMPK induced WAT browning by increasing UCP-1 levels that provoked energy expenditure using fatty acid as substrate. Over all of this study provided the evidence to support using selegiline as anti-obesity agent and its mechanism. However, there are some points that need more discussion. Then, I would like to send you some comments and would appreciate if you consider these suggestions as listed below.
1. Introduction line 56-58; As author mentioned that “MAO also accelerates adipogenesis by generating ROS (ref). Hence, the inhibition of MAO not only stimulates glucose uptake [13] but also inhibits lipolysis [14] in adipocytes of rats.”
- Please cite the reference of the first sentence.
- As ref. 13 & 14, the original articles showed the substrate of MAO stimulate glucose transporter (ref.13) and inhibit lipolysis (ref.14), therefore inhibition of MAO results in inhibited glucose uptake and induced lipolysis, please check this.
2. Introduction line 62; author mentioned that “there are previous study that have been described how selegiline reduces body wight gain and improves metabolic parameter in obese rodents.”
- Does selegiline reduce body weight gain and improve metabolic parameter through inhibition of MAO? Author should give more detail and cite the reference of original paper. It would support your idea that why you have to investigate the other mechanism of selegiline.
3. Method Session 2.3 line 109; Author mentioned Real-Time qPCR in method and in discussion, line 306 … the expression of adipogenic gene (C/EBP-306 ɑ, FABP4) ….. and line 324 … the mRNA and protein expression of UCP1…, but there are no any results of gene expression both in result section or figure.
4. Result Session 3.5; line 283 “We did note a significant difference in UCP1 expression levels in the BATs of the HFD-283 Veh and HFD+selegiline groups.” This sentence is discrepant with the sentence in discussion line 329 “….., there was no change in UCP1 expression in the BATs,…..”
5. Discussion line 311; “Phosphorylated HSL is then broken down into fatty acids within lipid droplets,….” HSL is an enzyme (protein) that catalyze the hydrolysis of triacylglycerol into FFAs and glycerol, thus itself cannot be broken to fatty acid.
6. Discussion;
- paragraph 3 and 4 should be discussed together in detail of AMPK signaling pathway involving lipid metabolism (downstream marker) and show how selegiline can alleviate obesity through AMPK activation. I encourage you to discuss with other research articles that show the same effect (anti-obesity by activated AMPK), it would be support you research work and reliable.
- As author mentioned at line 330 about AMPK and WAT browning, author should more detail discussion how AMPK related to browning, Dose it involve with UCP-1, PRDM16? Are they downstream of AMPK ? What is the importance of these protein.
Minor
1. The title ….HFD-fed obese mice would be better than mouse.
2. a selective MAO type B inhibitor should be mentioned after word of selegiline (introduction, line 59)
3. Method Session 2.1; Number of mice was mentioned in NCD group (n=4), How many mice in HFD group (line 83-84)? Please mention in the method.
3. Besides epididymal white adipose tissue, inguinal and brown adipose tissue were collected, please mention in the method.
4. Figure 1. Cohort2 as a treatment model, time line treatment should be same as Figure 3 that the treatment of selegiline start at week 16.
5. Figure legend of Figure 2.
Line 172 … mice fed NCD or HFD (n=5 per group), the number of mice is discrepant with the number in the method (session 2.1), please check it.
Line 175 unit of μm2, make super subscript of 2.
6. Figure 5.
- There is alphabet e in the image of WB result. What does it mean?
- The figure legend is discrepant with the image (A) Left panel: Images of Western blot of….. Right panel: Summary plot of expression …. , (B) Left panel: Images of Western blot…. Right panel: Summary plot of … , please check it.
7. Figure 6.
- The figure legend is discrepant with the image, Right panel: Summary plot of expression of PRDM16, PGC1ɑ, …. It is Figure B not Right panel. There are Figure A-E, please check the figure and the description.
- Figure B, figure legend of y axis is missing.
8. Result Session 3.5 Figure should be cited in text after result description for example line 279-281 “Treatment with selegiline was observed to have induced higher levels of UCP1 expression levels in the eWATs than in the untreated group (Fig.6 A-B). Line 282 “…. in the HFD-Veh group (Fig.6C)”
Comments on the Quality of English LanguageProofreading is required. The structure of some sentences must be rechecked.
Author Response

(The authors gave the same response as above.)

Reviewer 5 Report
Comments and Suggestions for Authors
Young et al. investigated the effects of selegiline on HFD-fed obese mice and characterized possible underlying mechanisms. The results are generally presented, with statistical analysis applied. The discussion integrates the results with existing literature, and potential mechanisms are proposed.
Below are my comments:
Introduction: Please make relevant introduction of selegiline.
Methods: Why there were 2 cohorts used? Please make an explanation here.
How was the dose of selegiline selected?
Fig. 2B: BW gain should be analyzed by 2-way ANOVA with repeat (day) measures. Same were seen in Fig. 3C and 3F.
Acronyms should be used correctly: Line 45 and 220, BAT; line 57, ROS; section 2.4, antibodies; etc...
Comments on the Quality of English LanguageEnglish is fine.
Author Response

(The authors gave the same response as above.)

Round 2
Reviewer 1 Report
Comments and Suggestions for Authors
The authors answered all questions in a relatively satisfactory way.
Comments on the Quality of English LanguageNot much concern.
Author Response
The reviewer's constructive advice helped us improve the content of our manuscript. We would like to thank the reviewer.
Reviewer 2 Report
Comments and Suggestions for Authors
Please double check the writing for better reading and unerstanding.
Author Response

(The authors gave the same response as above.)

Reviewer 3 Report
Comments and Suggestions for Authors
Revision is acceptable.
Author Response

(The authors gave the same response as above.)

Reviewer 4 Report
Comments and Suggestions for Authors
The author well response to the suggestions and comments. This revised version can now be accepted.
Author Response

(The authors gave the same response as above.)
